# Spectroscape enables real-time query and visualization of a spectral archive in proteomics

Long Wu [1,2], Ayman Hoque[1] & Henry Lam [1] ✉

In proteomics, spectral archives organize the enormous amounts of publicly available peptide tandem mass spectra by similarity, offering opportunities for error correction and novel discoveries. Here we adapt an indexing algorithm developed by Facebook for organizing online multimedia resources to tandem mass spectra and achieve practically instantaneous retrieval and clustering of approximate nearest neighbors in a large spectral archive. An interactive web-based graphical user interface enables the user to view a query spectrum in its clustered neighborhood, which facilitates contextual validation of peptide identifications and exploration of the dark proteome.

The tandem mass spectrum is the basic unit of proteomic data, each representing a direct experimental observation of a peptide, and by inference, the parent protein. A modern mass spectrometer can easily produce tens of thousands of MS/MS spectra every hour. Such rapid data generation, when combined with the increasing prevalence of these machines, has led to an astronomical amount of data. In ProteomeXchange, the leading data repository in proteomics, over 5300 datasets are deposited in 2021 alone, with the number of datasets deposited steadily increasing every year[1–3]. Within these datasets, the number of MS/MS spectra currently available in the public domain is on the order of tens of billions[2,3]. Moreover, this data volume probably represents only a small fraction of all proteomic data generated, as typically only datasets directly associated with publications are deposited. Currently, proteomics data are largely stored in data repositories, which organize the data by datasets associated with individual publications or projects[4–7]. In effect, the spectra are stored in the same unprocessed data files and organized in the same structure as when they were submitted, usually as supporting data of a publication. As many have argued, while this is an important first step, such organization is not conducive to meaningful data reuse[8,9].

The emerging paradigm explored by several proteomics databases is to organize data by spectral similarity, in what is called a spectral archive[10,11]. A spectral archive preserves all individual spectra, identified or not, and spectra similar to each other are grouped in a process called spectrum clustering. In theory, this organization should maximize the potential for data reuse. For instance, by analyzing clusters of similar spectra, the unidentified ones among them can become identified through association, errors in identifications can be corrected, outliers can be detected, and cross-references across datasets can be made. Therefore, once in a spectral archive, the identification of a given spectrum can evolve and hopefully self-correct over time[11–13]. Another application of spectral archives, which was demonstrated previously, was to discover unexpected post-translational modifications or amino acid substitutions by detecting mass shifts between connected members in a cluster[14]. However, it is computationally challenging to maintain large spectral archives, since in principle each spectrum needs to be compared to every other spectrum, though in practice the problem is made more tractable by various shortcuts. Some common strategies are: (i) progressively reducing the search space by merging spectra immediately in newly found clusters, (ii) limit comparisons to spectra with similar precursor m/z values, and (iii) avoiding comparisons of dissimilar spectra by grouping similar spectra first using various dimensionality reduction schemes[10,13,15–21].

Despite the obvious promise of spectral archives, the adoption of the idea by the wider proteomics community has been slow. In our view, the main reason is that spectrum clustering is computationally intensive, and does not seem accessible to the typical proteomics researcher. Moreover, the outcome of the spectrum clustering is not visible to the human user, and must be further analyzed by other computational tools, as existing tools lack a graphical user interface that allow the user to interact with the spectral archive.

[1]Department of Chemical and Biological Engineering, The Hong Kong University of Science and Technology, Clear Water Bay, Hong Kong. [2]Department of Electrical and Computer Engineering, The Hong Kong University of Science and Technology, Clear Water Bay, Hong Kong. ✉e-mail: kehlam@ust.hk

To break this inertia, we propose Spectroscape, a platform that enables the real-time query and visualization of a spectral archive (Fig. 1). Unlike other spectrum clustering tools that focus on pre-building the spectral archive offline, our method enables query-driven real-time clustering, providing the detailed cluster structures with connectivity information. This is made possible by an indexing scheme based on the inverted file and product quantization encoding (IVF-PQ) algorithm in the Facebook AI Similarity Search (FAISS)[22] library that groups spectra in neighborhoods in high-dimensional space, defined by approximate spectral similarity. Given any query spectrum that the user supplies, the method efficiently retrieves all its approximate nearest neighbors in the entire repository and performs real-time spectrum clustering by computing accurate pairwise distances among the query and the neighbors to reveal any cluster(s) in the neighborhood. The user can then visualize the result in an interactive web-based user interface. In effect, the platform allows the user to search the entire data repository by spectral similarity, and instantaneously receive not only a list of best-matching spectra to the query, but also the detailed structure of any cluster in its neighborhood. Spectroscape is fast enough even without limiting comparisons by using the precursor m/z value, making it possible for the user to discover unexpected post-translational modifications or sequence variants by inspecting clusters of spectra of different precursor masses[23,24].

## Results

In Spectroscape, a spectrum is represented by a vector in 4096-dimensional space. The chosen variant of the IVF-PQ algorithm converts such a vector into a 17-byte string that serves as an approximate

address of its location. In brief, the 4096-dimensional space is first divided into 256 regions (called buckets) in a training step that takes in a random sample of MS/MS spectra and uses the familiar k-means clustering algorithm[25] to partition them into 256 groups. Then, the residual vector, the difference between the spectral vector and the centroid of the bucket it belongs to, is sub-divided into 16 vectors, and for each of them, a similar process is carried out to map it to one of 256 possible sub-buckets. Thus, for every spectrum, its address consists of two parts: a bucket number, which denotes which of the 256 buckets it belongs to, and the 16 sub-bucket numbers of its residual vector. The beauty of this indexing approach lies in the efficient computation of distances between any two addresses (Supplementary Method 1). Although the address is only an approximation of the actual location of the spectral vector, this method enables the efficient retrieval of top approximate nearest neighbors (ANNs) of any query spectrum. With suitable parameters, which were optimized in this study (Supplementary Method 2 and Supplementary Fig. 1), the probability of recovering the true nearest neighbors among the ANNs is very high. It then remains to perform detailed spectrum clustering among the approximate nearest neighbors to reveal the full cluster structure in the neighborhood of the query spectrum, but since the number of retrieved ANNs is capped, the time taken for detailed clustering would not scale with the archive size. It should be noted that unlike some existing spectrum clustering tools, which attempt to cluster similar spectra as they are loaded into the archive, Spectroscape only organizes spectra loosely in buckets with pre-defined boundaries each containing millions of spectra during the indexing step. The retrieval and the detailed clustering of the ANNs are only performed

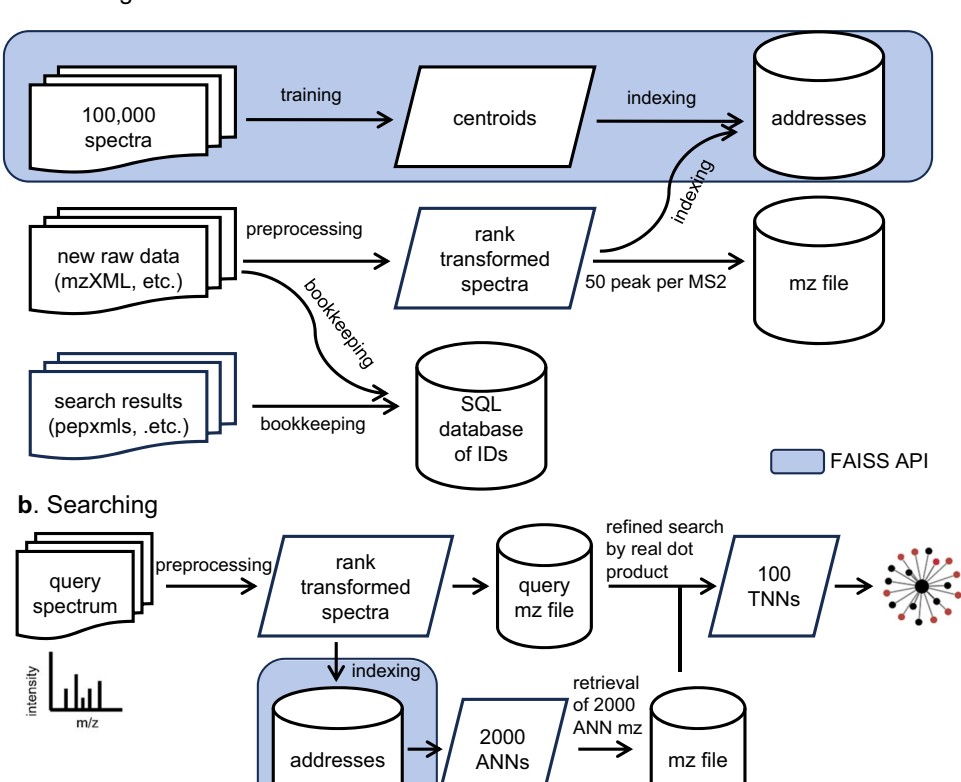

**Fig. 1 | Building and searching of the spectral archive by Spectroscape.** In the building step (**a**), the indices are built by training on 100,000 spectra. The new data files can be added to the indices after preprocessing. The preprocessed spectra are also stored in an MZ file. An SQL database is created to keep all the meta information

of each spectrum, including identifications (if any). In the searching step (**b**), ANNs are retrieved from indices for each preprocessed query spectrum. Those ANNs are re-ranked by the accurate dot product to retain up to 100 true nearest neighbors (TNNs) for clustering, which is visualized as a force-directed graph.

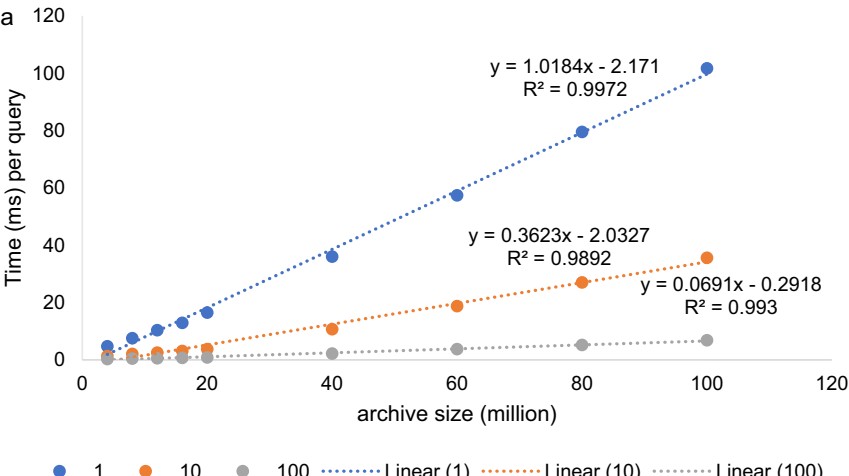

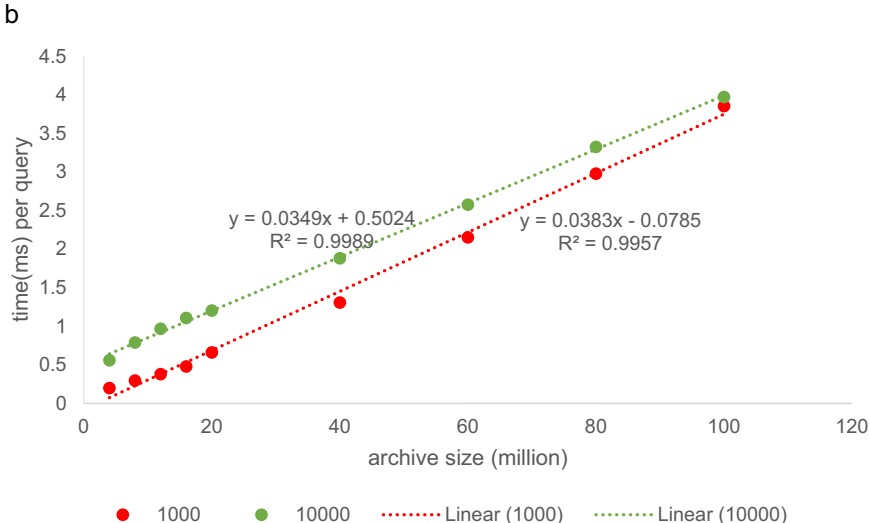

**Fig. 2 | The average time of ANN retrieval.** The average time for retrieval of ANNs for each query spectrum on one IVF-PQ index, as a function of the number of spectra in the archive, in millions, if one query (**a**, blue), 10 queries (**a**, orange), 100 queries (**a**, gray), 1000 queries (**b**, green) and 10,000 queries (**b**, red), respectively, were executed in a batch. As shown, the per-query retrieval time scales linearly with archive size. All the five cases result in a $R^2$ around 0.99 in linear regerssion. The running times are obtained on a 32-core AMD Ryzen Threadripper PRO 3975WX CPU.

on-demand at query time to reveal the neighborhood of the query spectrum. This means that newly acquired spectra can be added to the archive quickly, without comparing them to the existing spectra, which is essential for the incremental update of the archive and hence scalability.

We have tested the method on about 106 million spectra of several large datasets of human samples and verified that the query time scales linearly with archive size (Fig. 2). One individual query can be executed in about 0.1 s. The per-query search time can be further reduced to less than 5 ms if many queries are processed in a batch, thanks to amortization of the computational overhead. This is achieved on our modest server with a 32-core CPU. Spectroscape can also be run on graphical processing units (GPUs) for further parallelization, which is supported by the FAISS library, provided that the GPU has enough internal memory. In our limited testing of a smaller archive of 25 million spectra, a fivefold further reduction in search time can be achieved with one entry-level GPU.

The success of the indexing scheme is best assessed by the recall, defined as the probability of finding the true nearest neighbors (TNNs) among the retrieved approximate nearest neighbors. By testing a random sample of 20,000 query spectra searched against a spectral archive with 106 million spectra, we found the overall recall is over

98%. For 94% of the queries, perfect (100%) per-query recall is achieved (Fig. 3). Here, we define the TNNs as the N most similar spectra ($N \leq 10$) to the query in the entire repository, with the condition that the spectral dot product is at least 0.7, a typical cutoff for a good spectral match. The overall recall would be lower at 96% (with 83% of queries achieving perfect recall) if we require the retrieval of up to the 100 most similar spectra with dot product at least 0.7. If we relax the dot product threshold to include fainter matches as TNNs, the recall is slightly reduced, while the opposite is true if we raise the threshold, indicating that the retrieval accuracy of the algorithm is higher if the neighbors are more similar. Fortunately, this limitation, shared by many ANN-based algorithms, is irrelevant for our intended application, as there is no information gained by finding such far-away neighbors, essentially random spectral matches. The high recall can be attributed to the ability of the IVF-PQ algorithm of Spectroscape to approximate the true spectral similarity between any two spectra by the dot product between their respective 17-byte addresses. Figure 4 shows the high correlation between the true dot product and the approximate dot product by IVF-PQ, whether or not the two spectra have similar precursor masses. For comparison, the same spectra were embedded in 32-dimensional vectors by the deep learning-based clustering tool, GLEAMS (Supplementary Method 3). The correlation between the true

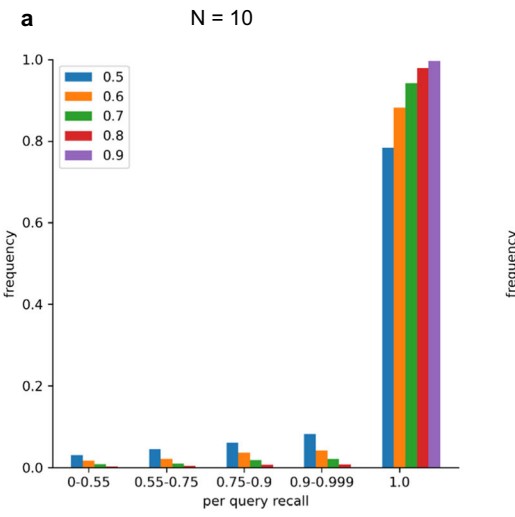
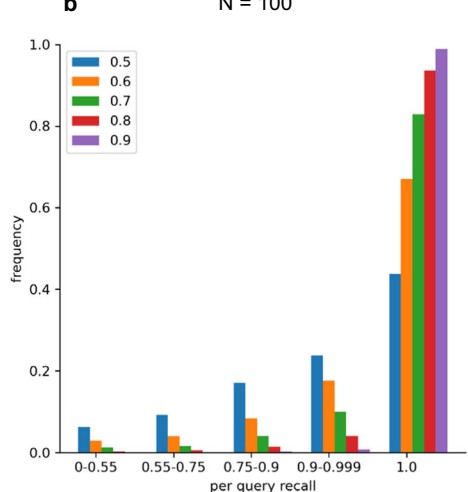

**Fig. 3 | Recall of TNNs by Spectroscape against a spectral archive with 106 million spectra.** Histograms of per-query recall of the TNNs, tested with 20,000 randomly selected queries from the archive. TNNs are defined by two parameters: $N$, the number of closest neighbors to be retained as TNNs ($N = 10$ and 100), and $t$, the minimum dot product to be considered a TNN ($t = 0.5$–$0.9$). On both (**a**) ($N = 10$)

and panel (**b**) ($N = 100$), the per-query recall distributions are dominated by the rightmost column, representing the fraction of queries achieving perfect recall. In general, Spectroscape achieves higher per-query recall if one requires higher minimum dot product similarity to be counted as TNNs, or if one considers fewer TNNs.

dot product and the dot product of the GLEAMS embeddings was clearly worse, even though the embeddings (32 floating point numbers) contain far more information than the IVF-PQ addresses (17 bytes). Moreover, in GLEAMS, the correlation between the true dot product and the approximate dot product (the dot product of the embeddings) drops significantly from 0.6 to 0.3 if the pair of spectra are of different precursor masses. This can be explained by the fact that, unlike Spectroscape, GLEAMS uses the precursor mass or peptide identification to train the embedding. Taken together, this suggests that the Spectroscape IVF-PQ addresses, despite its simpler implementation and smaller memory footprint, can preserve the true spectral similarity better than GLEAMS embedding. More importantly, Spectroscape, being agnostic to precursor information, is much better positioned to discover mass-shifted spectral matches and detect potential PTMs and sequence variants.

Under our optimized default settings, about 2000 distinct ANNs are retrieved for a given query by the index. For the purpose of visualization, Spectroscape re-ranks the ANNs by accurate dot products to obtain the list of TNNs (Supplementary Method 4). Then it performs true pairwise similarity calculations of the retained TNNs to determine the cluster structure to be displayed in a graph. This occurs in real time without any noticeable lag time. Each node of the graph is a spectrum, with the query positioned in the center, and two nodes are joined if their true similarity is above a certain user-defined threshold. Hence, groups of highly similar spectra will tend to form tightly-connected clusters. The node is color-coded based on the peptide identification of the spectrum, or colored black if it is unidentified. The edge is colored black if it connects spectra of similar precursor $m/z$ values, and orange if the two spectra are not similar in precursor $m/z$. This enables the user to quickly detect discordant identifications among cluster members, as well as mass-shifted spectral matches indicative of PTMs or sequence variants. The edges connecting two nodes with dissimilar $m/z$ or mass value can be removed by toggle an option by the user on the web interface.

To provide a visual impression of the tightness of the cluster, a force-directed graph drawing scheme was employed (Supplementary Method 5), such that the edge lengths are not a simple function of the spectral similarity, but also depend on the cluster structure. (An option exists to turn off this behavior and set the edge length to correlate with the true Euclidean distance.) The force-directed graph drawing is

especially conducive for detecting loosely connected subclusters and highlighting outlying and bridge nodes. This feature can have useful applications. For one, chimeric spectra can be readily detected, as they tend to bridge two clusters of different identifications. For another, the subclusters can reveal subtle differences in fragmentation patterns of the same peptide ion, perhaps due to instrument differences. Some examples of how the visual cues provided by Spectroscape can help confirm identifications or lead to new discoveries are shown in Fig. 5 and Supplementary Fig. 2.

Another interesting feature of this platform is that consensus spectra in spectral libraries can also be loaded into the archive and be retrieved along with the experimental spectra. In effect, a conventional spectral library search is carried out as part of the query, without limiting to candidates of similar precursor $m/z$ values. (A demonstration of this potential application is described in Supplementary Note 1.) In most cases, the consensus spectrum tends to be located near the cluster center comprising its experimental replicates, which is expected since the consensus spectrum is a weighted average of the replicates. However, a given query is often more similar to some experimental replicates than it is to the consensus. By its nature, the consensus construction process distills only the most reproducible features of the peptide fragmentation pattern, and may throw away partially reproducible features that can also be used to support some spectral matches[26]. Occasionally, it can be seen that the consensus spectrum resides at the outer fringe of the cluster, suggesting that it may not be truly representative of the experimental replicates. Figure 5c offers such an example, in contrast to the ideal scenario shown in Supplementary Fig. 2a. Therefore, with the ability to query the entire spectral repository, we envisage a more accurate and sensitive method for spectral identification in the future that bypasses the step of model generation (i.e., construction of the consensus spectrum) from observations, but relies on matching experimental observations directly and in a contextual manner.

The web-based interface also enables the user to interact with the spectral archive in real time. For example, the user can search the archive with any displayed node as a new query by double-clicking it, thereby hopping from node to node as he/she navigates the neighborhood for more insights. It also provides many options for the user to customize how the spectrum clusters are displayed, by changing the sizes and colors of the nodes and edges, and tuning the parameters of

**Fig. 4 | Approximation of true spectral similarity by Spectroscape and GLEAMS[19].** The scatterplots of true dot products versus the approximate dot products (Spectroscape, left pane; GLEAMS, right pane) for: **a** all pair of spectra, **b** pairs of spectra with similar precursor mass (within 3 Da), and **c** pairs of spectra with dissimilar precursor masses. Each data point represents one retrieved ANN of one of 20,000 queries randomly chosen among the spectra in the archive. The color indicates density of data points by kernel density estimation. The Pearson correlation (r) is shown on top of the scatterplots. The correlation between the approximate dot product of Spectroscape and true dot product is above 0.8 when considering all the spectrum pairs or the pairs with similar precursor masses. This correlation coefficients drops slightly to 0.7 when considering only spectrum pairs with dissimilar precursor masses. The correlation between approximate dot product of GLEAMS and true dot product is around 0.6 when considering all the spectrum pairs or the spectrum pairs with similar precursor masses. The correlation drops significantly to 0.3 for pairs with dissimilar precursor masses, suggesting that the trained embedding of GLEAMS relies heavily on precursor information.

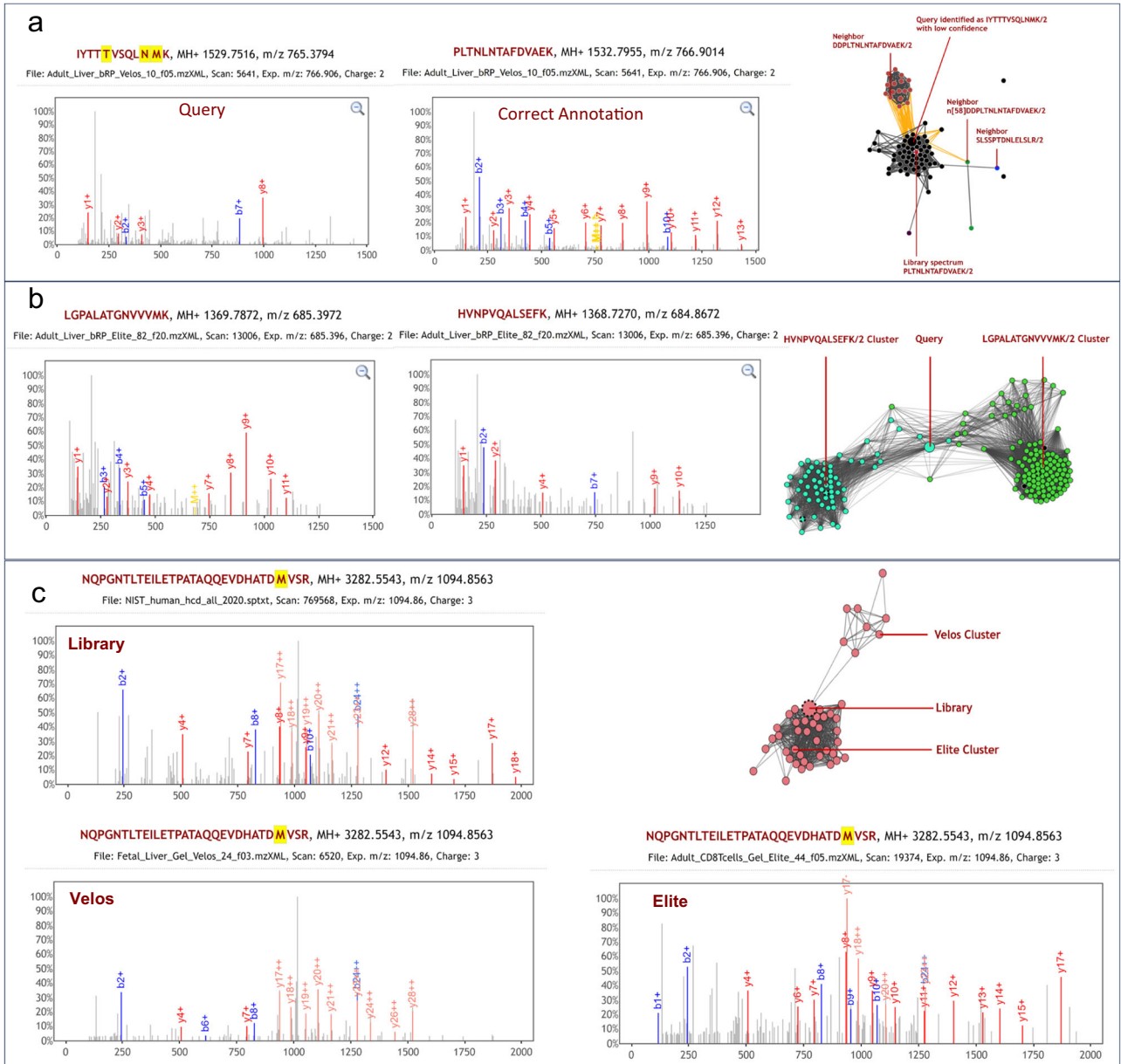

**Fig. 5 | Examples of spectrum clusters visualized by Spectroscape.**
**a** Identification of unidentified spectra by association with neighbors with a precursor mass shift. The query node (PSM shown in top left panel) is unidentified but its nearest neighbors, with a precursor mass shift of 230.04 Da, are identified as DDPLTNLNTAFDVAEK/2. The mass shift equals to the mass of the oligopeptide DD. Therefore, the unidentified query node and all its black neighbors are likely the semi-tryptic peptide PLTNLNTAFDVAEK/2, which can also be confirmed by the fact that a library spectrum of that peptide ion is located in the middle of that cluster. The bottom left panel shows a very good PSM when we re-annotate the query node with this semi-tryptic peptide. **b** A query spectrum as a bridge of two clusters of PSMs corresponding to two peptides, LGPALATGNVVVMK/2 and HVNPVQALSEFK/2. It is clearly a chimeric spectrum as the two PSMs almost do not share any matched peaks, as shown by the same spectrum annotated with respect to the two peptide ions (left panel). **c** The same peptide ion NQPGNTLTEILETPATAQQEVDHATDM[147]VSR/3 yields two slightly different fragmentation patterns on two mass spectrometers, as can be visualized clearly as two subclusters. The exact reason behind the difference in fragmentation patterns is unclear, but may be a combination of differences in instruments (including their conditions at the time), experimental parameters, and local sample complexity at the time of ion selection and fragmentation. The NIST HCD library spectrum (top left), represented as the node with a dotted outline that lies in the middle of the two subclusters, does not capture either of the fragmentation patterns very well.

the force-directed graph. The interface also shows the full spectrum of any node in an interactive spectrum viewer, or a butterfly plot of the spectra of two selected nodes shown head-to-tail for comparison. A full table of nodes and edges containing their information is available in a separate tab. Some screenshots of the Spectroscape interface are shown in Supplementary Figs. 3–7.

In terms of applications, Spectroscape is enables users in proteomics to assess the plausibility of any peptide-spectrum match (PSM) in the context of prior data, and do so in a visual and interactive

manner. A user will be visually prompted to question a PSM that contradicts the identifications of vast majority of its nearest neighbors. This is in contrast to the current practice of relying on search engines and statistical validation tools that operate on the data at hand, which do not take advantage of the huge amount of publicly available data nor provide alternative explanations. With this platform, unexpected PTMs, sequence variants and chimeric spectra can also be readily discovered, which sometimes offer better interpretations than the answers found by the search engine. Although we tested Spectroscape

on human proteome data only, one can imagine a combined multi-species spectral archive being useful for detecting sequence homologs between species.

Spectroscape complements state-of-the-art methods for statistical validation that seek to control the global false discovery rate (FDR) at the file or dataset level in an unsupervised manner. Although such automatic validation is convenient and necessary as a first-pass filter for potential discoveries from high-throughput data, it is fundamentally inadequate if any biological conclusion critically depends on a few PSMs, because the globally estimated FDR cannot be taken to apply to any small subset of the retained PSMs[27,28]. This shortcoming poses a challenge in confirming the discovery of novel rare proteoforms, for which careful examination of such PSMs by a human expert is still necessary. At present, journal editors and reviewers are sometimes tasked to assess the scientific merits of such discoveries, but often with limited means to view the key spectra in question, let alone seeing them in context. Spectroscape can supplement recent efforts to facilitate and standardize data reporting at the spectrum level[29] and help address this unmet need.

Spectroscape is specifically designed for visualization. It is geared towards on-demand, detailed clustering of a small neighborhood to be inspected manually, instead of clustering an entire archive in a batch operation as is done by existing clustering tools. This design choice means that most of the computation happens at query time rather than loading time. Therefore, adding new data to the archive incrementally is straightforward, as each new spectrum is simply mapped to a 17-byte address and loosely sorted into one of 256 buckets in preparation of ANN retrieval. No re-training of the indices or clustering with existing data is performed at loading time. On the other hand, because only a small neighborhood is visualized on prompt, one can afford expensive pairwise similarity calculations among the neighbors to reveal the cluster structure, which greatly facilitates interpretation. Moreover, although Spectroscape is designed for on-demand visualization, one may take advantage of its speed for offline spectrum clustering. An illustrative application of using Spectroscape to re-annotate unidentified or potentially mis-identified spectra automatically through cluster membership is presented in Supplementary Note 2.

Finally, we envision that Spectroscape can eventually be deployed as an integrated platform for both data analysis and data sharing in public repositories, as the query process and the spectral archive-building process are algorithmically the same. Any new data submitted can be quickly indexed, and become part of the spectral archive and connected to prior knowledge. Once the indexing is complete, the data submitter then receives a notification with links to browse his/her data through Spectroscape. This type of workflow is nothing new in genomics, as sequencing data is almost always analyzed in the context of prior data thanks to convenient tools like BLAST, and through that process, any new data deposited into public databases are quickly integrated and become useful to everyone. With the concerted efforts of the proteomics community, we believe that this platform can grow and evolve to play a similar role for proteomics.

## Methods

### Datasets

Two large human proteome datasets were included in the spectral archive, both downloaded from ProteomeXchange[30]. The first one, accession PXD000561, consists of 2210 LC-MS runs on Thermo Scientific LTQ Orbitrap Velos and LTQ Orbitrap Elite mass spectrometers[31]. The second one, accession PXD010154, consists of 1796 runs on Thermo Scientific Q Exactive Plus, Q Exactive HF, and Orbitrap Fusion Lumos mass spectrometers[32]. The raw data files were converted into mzXML format using MSConvert[33]. The resulting 106,196,959 MS2 spectra, all high-resolution HCD spectra albeit from different instruments, were searched with MSFragger (v20190628)[34]

with search parameter as follows. Precursor and fragment mass tolerance are 0.05 Da and up to 2 missed cleavages are allowed. One fixed modification (cysteine carbamidomethylation), and 7 variable modifications (methionine oxidation, protein n-terminus acetylation, asparagine deamidation, peptide n-terminus carbamidomethylation, threonine carbamidomethylation, and tryptophan dioxidation) were considered. The sequence database of human proteome is downloaded from Uniprot[35] with accession id UP000005640. Decoy database is created with in-house Perl script, where the amino acid sequence within a tryptic peptide is shuffled while keeping the K and R fixed. The search results were processed by PeptideProphet and iProphet in the Trans Proteomic Pipeline[36] (TPP v6.0.0 OmegaBlock). All spectra regardless of identification confidence were loaded into the spectral archive; the iProphet[37] probability was used only to determine if the corresponding node is colored when displayed. Three consensus spectral libraries containing over 900,000 HCD spectra in total from human samples compiled by the National Institute of Standards and Technology (NIST), USA, was downloaded from https://chemdata.nist.gov/dokuwiki/doku.php?id=peptidew:lib:humanhcd20160503 (dated 2020-05-19), converted to sptxt format by SpectraST[26], and loaded into the archive.

### Spectrum preprocessing

All spectra are preprocessed as follows. First, spectra with number of peaks lower than a user-defined threshold (6 by default), are recorded as an empty spectrum. Next, only the top 50 most intense peaks for each spectrum are retained, and the intensity is rank-transformed to a scale of 50 (most intense), 49, 48, ..., 2, 1. To maximize the information content, isotopic peaks and peaks near the precursor $m/z$ were excluded, and no more than 6 peaks within a sliding window of 50 $m/z$ are kept[38]. Then the peak list is vectorized by binning, with a bin width of 0.5 $m/z$, resulting in a vector of 4096 dimensions, the required input to the IVF-PQ algorithm. In the binning step, the two flanking bins of each peak are filled with half of the rank-transform intensity.

### Training and optimizing the index

The FAISS IVF-PQ index needs to be initialized in a training step. A randomly selected set of 100,000 spectra was used to find an optimal partition of the high-dimensional space that distribute the spectra among buckets and sub-buckets. The details of the algorithm were described in ref. 22 and briefly outlined in Supplementary Method 1. The process takes about 15 min on our server. This process can be repeated using a different set of training spectra and different product quantization schemes[39] to produce distinct indices. At the expense of running time, combined use of multiple indices can improve recall, because pairs of similar spectra that straddle a bucket boundary of one index are unlikely to do the same in another index. Thus the number of indices used (*nindices*) is a parameter that can be chosen to strike a balance between recall and running time. Moreover, at the time of query, one can limit the number of closest buckets (*nprobes*) to search for approximate nearest neighbors, which is another parameter that can be chosen. After optimization, we chose the parameter settings of *nindices* = 2 and *nprobes* = 8 for the rest of the study (Supplementary Method 2 and Supplementary Fig. 1), though the latter can be changed at query time by the user in the web interface of Spectroscape.

### Adding data to the spectral archive

Tandem mass spectra in MS data files (in mzXML, mzML, mgf or sptxt formats) are first preprocessed to retain top 50 peaks, and the 17-byte address of each spectrum is computed by the IVF-PQ algorithm for each index. Meanwhile, to enable the subsequent clustering and visualization steps, the preprocessed spectrum is stored in a memory-efficient format in a separate file (called the MZ file) whereby the m/z values of the retained 50 peaks were converted to a two-byte integer

and listed in descending order of intensity, resulting in a 100-byte representation. Other metadata of the spectrum such as the source data file name and scan number, its precursor m/z value, and its peptide identification (if any, loaded separately from the corresponding pepXML[40] or tab-delimited file containing search engine results) is stored in an SQLite database[41] in the bookkeeping step (Fig. 1a). The corresponding entries in the indices, the MZ file and the SQLite database are linked by a unique spectrum ID.

### Retrieval by spectral similarity

A query request can be sent through a web browser to the server, which executes the query in a cgi script written in C++. The user can either use one spectrum already loaded into the spectral archive as the query (by providing the unique spectrum ID, or by searching by the source data file name and the scan number, or the peptide identification), or upload a new query spectrum to the server through the web service. Searching a batch of queries is also possible on the server side via command-line scripts.

Given a new query spectrum, the same preprocessing steps are taken and its address (one per index) is computed. For each index, the IVF-PQ algorithm then retrieves all the existing addresses in the closest *nprobes* buckets, and computes the approximate dot product between each address to the query's address (Supplementary Method 1). The top 1024 most similar addresses are returned as the approximate nearest neighbors (ANNs) of the query. The ANNs returned by all indices are combined, and the corresponding preprocessed spectra are read from the MZ files. Then, the accurate dot product is calculated between the full query spectrum and the full spectra of the ANNs to determine the true nearest neighbors (TNNs).

### Clustering and visualization

For visualization, the N most similar TNNs (N defaults to 20 but can be chosen by the user) can be shown in a graph in Spectroscape, with each spectrum shown as a node. For these N spectra, pairwise accurate dot products are computed to get a fully connected graph. For visualization in the web browser, two spectra are connected by an edge if their Euclidian distance exceeds a user-defined threshold. The graph is then plotted by the web browser using a force-directed graph drawing algorithm implemented in a free JavaScript library D3.js (https://d3js.org/). In brief, this graph visualization simulates the physical behavior of a network of electrically charged spheres (the nodes) connected by springs (the edges), with several parameters controllable by the user (Supplementary Method 5). The query spectrum is displayed in the center of the graph as a larger circle to distinguish it from its neighbors. The color of the node is calculated based on the peptide sequence using a hash function to map a string to a RGB color. The color black was reserved to indicate an unidentified spectrum. Optionally, edges between spectra of different precursor m/z values can be shown as a different color, to aid the discovery of potential PTMs and variants. Information about the spectra and their computed similarities are shown if the mouse pointer hovers over nodes and edges. Single clicking on any node displays the full spectrum by the Lorikeet spectrum viewer. If the spectrum is identified by search engine with FDR < 1%, it will be annotated with expected ions. To help the user evaluate alternative explanations for the query node, the user can also enter another peptide identification, upon which the annotation will be updated. Double-clicking on any node will start a new query with the clicked node as the query spectrum.

### Running time evaluation

Spectroscape was tested on a server equipped with an AMD Ryzen Threadripper PRO 3975WX CPU with 32 cores at 3.5 GHz, one NVIDIA RTX3080Ti 12 GB GPU, and 512 GB of DDR4 3200 MHz RAM. Training two IVF-PQ indices using 100,000 spectra took about 15 min in total. Adding one spectrum to the archive, including all the preprocessing, indexing and bookkeeping steps, took about

0.37 ms on average. To estimate retrieval time, different query batch sizes (1, 10, 100, 1000 and 10,000) were tested; a random sample of spectra already in the spectral archive are submitted for query to evaluate the effect of parallelization. The running times for various steps done on the server were recorded by the commands in the C++ standard library chrono, and averaged over many executions. The time taken for sending query requests over the internet, or for plotting the graph in the web browser is not estimated given the inherent variability of such tasks.

### Recall evaluation

To evaluate the retrieval performance of the IVF-PQ algorithm, we measure the recall $R_i$ of a given query $\mathbf{q_i}$ by:

$$R_i(t) = \frac{|A(\mathbf{q_i}) \cap T(\mathbf{q_i}, N, t)|}{|T(\mathbf{q_i}, N, t)|} \quad (1)$$

where $A(\mathbf{q_i})$ and $T(\mathbf{q_i}, N, t)$ is the sets of retrieved ANNs of $\mathbf{q_i}$ and its TNNs, respectively, and $t$ is the minimum dot product threshold to retain as a TNN. Here, the set of a TNNs of a given query $T(\mathbf{q_i}, N, t)$ is obtained by comparing the query spectrum to every spectrum in the archive without any index, and retaining up to $N$ nearest neighbors whose accurate dot products with the query are above a certain cutoff $t$. The recall $R_i$ of 20,000 randomly selected queries searched against a spectral archive with 100 million spectra were plotted in two histograms corresponding to top 10 and top 100 TNNs (Fig. 3a). The dot product cutoff was varied between 0.5 and 0.9, to evaluate the impact of true spectral similarity on the recall, as it is expected that the ANN-based similarity retrieval should be more effective as the neighbors are closer in reality. The approximate dot products and the true dot products of all retrieved ANNs of one typical query are plotted in a scatterplot (Fig. 3b). The overall recall is defined over all the 20,000 queries as follows.

$$R_{overall} = \frac{\sum_{i=1}^{20000} |A(\mathbf{q_i}) \cap T(\mathbf{q_i}, N, t)|}{\sum_{i=1}^{20000} |T(\mathbf{q_i}, N, t)|} \quad (2)$$

### Reporting summary

Further information on research design is available in the Nature Portfolio Reporting Summary linked to this article.

## Data availability

The data used for building the spectral archive are available from ProteomeXchange repository with identifiers PXD000561 and PXD010154. The human sequence database was downloaded from UniProt with accession id UP000005640 (dated 2021-09-06). The human NIST libraries were downloaded from NIST [https://chemdata.nist.gov/dokuwiki/doku.php?id=peptidew:lib:humanhcd20160503] (dated 2020-05-19). The web-based interface to explore the spectral archive built on this dataset can be accessed from http://spectroscape.cc/. Source data are provided with this paper.

## Code availability

Spectroscape is implemented in C++ and JavaScript and is freely available as open-source software under MIT license at https://github.com/wulongict/SpectralArchive/[42] for individual groups to build their own spectral archive. The following third-party open-source packages were included in the Spectroscape distribution: FAISS v1.7.3, Boost v1.65.1, gtest v1.7.0, rapidxml v1.13, spdlog v1.x, eigen v3.3.1, MSToolkit (disseminated with comet v2016), and SpectraST v5.0. Nginx v1.18.0 and spawn-fcgi v1.6.4 are used for the web service of Spectroscape. D3.js v4.0, jQuery v3.3.1 and bootstrap v4.1.1 are in the web interface. Detailed installation and running instructions are provided in the README file.

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

## Acknowledgements

This work is supported financially by the Research Grants Council (Grant Nos. 16307620, 16306919, R5013-19, R4012-18, C6021-19EF and HKPFS for A.H.) the Innovation and Technology Commission (Grant. No. MHP/033/20) of the Hong Kong S.A.R., and the Hetao Shenzhen-Hong Kong Science and Technology Innovation Cooperation Zone project (Grant No. HZQB-KCZYB-2020083).

## Author contributions

L.W. developed Spectroscape and its user interface. L.W. created the archives and performed most of the data analysis. A.H. conducted illustrative computational experiments showcasing the functionalities of Spectroscape. H.L. conceived and supervised the project, and guided the algorithm development and data presentation. All authors contributed to the writing of the manuscript.

## Competing interests

The authors declare no competing interests.
