## [Peer Review File · Nature Communications]

REVIEWER COMMENTS

Reviewer #1 (Remarks to the Author):

In this revised version of the manuscript the authors have in my view addressed my previous comments: (i) they provide a larger benchmarking study (trying to get closer to a repository size dataset); (ii) they have improved documentation, including videos, and (iii) have tried to compare the performance of the tool with other previously published approaches.

I also find that the manuscript has now improved quite a lot in terms of scope and clarifying some aspects that maybe were not completely clear to me after my first review.

I agree with the authors that Spectroscape can indeed be a useful tool, but its real future impact cannot really be assessed at this point. A possible collaboration with a proteomics repository would also be key for the future although of course this is outside the current scope of this work.

Reviewer #2 (Remarks to the Author):

Wu et al. present an updated version of their manuscript presenting Spectroscape as a novel visualization tool for spectral archives.

The authors strongly argue, that their tool must not be viewed as another spectrum clustering algorithm, but a tool to investigate individual spectra.

I believe, that this argument falls short as it is necessary to show that the used method to group spectra compares to existing methods. Nevertheless, I follow the authors argument for this review.

I am surprised to see that all benchmarks etc were put in the supplementary methods. Therefore, the manuscript itself holds very limited information that would be relevant to a wider audience.

1) Novelty

The idea to create a visualization tool on top of a spectrum grouping / ANN network is nice, but not novel. This was done as part of previous PRIDE associated resources (although static and not scalable).

The methods used by the authors also were introduced by others before. Adapting the training process is innovative, but something I consider incremental improvement. Therefore, overall, the novelty of the proposed work is in my opinion limited.

2) Software

The state of the software was improved but is still below expectations. The installation works. Nevertheless, building an archive with only one 750 MB mzXML file failed with a segmentation fault on a 16 core machine with 32 GB RAM (again using a docker container and ubuntu:22.04). The mzXML file was taken from PXD016673.

The website to test the live version at <http://omics.ust.hk:8709/index.html> was not accessible during the time of the review (tested from multiple networks).

Overall, I therefore have severe reservations whether this tool will provide the simple and high usage of spectrum clustering or similar methods. It also raises grave concerns whether the authors hold the required expertise to develop an end-user ready piece of software and maintain it for a relevant amount of time. The original version contained severe and obvious errors and highlighted that the software was not tested properly. Apparently, the software still has not reached a stable version.

3) Theoretical usability

The tool (if it works) will only serve one question in proteomics research, to validate specific PSMs. The authors are correct, that some findings center on specific proteoforms / PTMs. Nevertheless, a vast number of literature centers on large-scale characterisations of proteomics samples. These will not profit from this tool.

As a mere proof of concept, this work is interesting, but not to the wide community addressed by this Journal. I therefore suggest this work to be submitted to a field-specific one.

Reviewer #3 (Remarks to the Author):

The authors have clarified the scope of the current work and have offered a compelling defense of the current capabilities of Spectroscape. There will be researchers looking at novel proteoforms or other non-standard identifications, and Spectroscape provides a unique tool for quickly exploring possible hypotheses regarding these discoveries. Additionally, the new analyses provided by the authors go a long way towards suggesting future developments for the technology.

In their rebuttal, the authors offered the following argument in defense of Spectroscape:

We believe that as a field we have gone too far down the road of controlling errors at a global level (e.g., by estimating FDR by target-decoy searching), while side-stepping the more fundamental question of what makes an individual PSM believable in the first place. Automatic “black box” approach to validation appeals to researchers who would rather focus on the biology, but we pay for this convenience with our credibility in the eyes of other life scientists. In a perfect world, the global error control we do today should only be viewed as the first step to filter away unpromising leads, but we should not be content to accept everything that passes the filter as truth. In just about every other discipline, scientists would not embark on expensive independent validation experiments until technical validation is done first and the data is deemed trustworthy. In proteomics, many researchers do not bother to even look at a spectrum that underpins our hypothesis, but are all too willing to embark on expensive and tedious biological validation experiments. With the right tool, checking even hundreds of PSMs perhaps take a matter of hours; in contrast, biological validation experiments often take months.

I would suggest adding something along these lines to the main manuscript. The argument against global FDR control is not new. Many papers have been written about the differences between global

FDR and local FDR or the False Positive Rate of a spectrum. For example,
<https://www.ncbi.nlm.nih.gov/pmc/articles/PMC2689316/>

These issues take on added importance when discovering novel proteoforms or searching non-standard databases. Even simple phospho enrichment experiments typically have different empirical FDR rates when subsetting the data to modifications on tyrosine. Referencing the known failures of global FDR control would bolster the argument in favor of using Spectroscopy as a validation step.

Regarding the main manuscript, my only new suggestion is that it would be nice for references to the Supplement to include a specific section identifier. In the current form it is somewhat inconvenient to match the main text with the relevant section of the supplement.

REVIEWER COMMENTS

Reviewer #1 (Remarks to the Author):

In this revised version of the manuscript the authors have in my view addressed my previous comments: (i) they provide a larger benchmarking study (trying to get closer to a repository size dataset); (ii) they have improved documentation, including videos, and (iii) have tried to compare the performance of the tool with other previously published approaches.

I also find that the manuscript has now improved quite a lot in terms of scope and clarifying some aspects that maybe were not completely clear to me after my first review.

I agree with the authors that Spectroscape can indeed be a useful tool, but its real future impact cannot really be assessed at this point. A possible collaboration with a proteomics repository would also be key for the future although of course this is outside the current scope of this work.

Response:

Thank you for the supportive comments on our manuscript. Yes, we are planning to make Spectroscape available over PeptideAtlas as a first step, and have been in touch with the engineers at ISB about this. However, it will take us months to sort out the technical (in particular hardware) issues within the ISB infrastructure, and we feel that we should not delay the dissemination of this work any longer. We hope that the Spectroscape tool can be easily accessible to everyone in the future, including researchers who want to build their own specialized spectral archives. Recently, we have also purchased a domain name, spectroscopy.cc to host a demo spectral archive persistently to promote the idea to the community.

Reviewer #2 (Remarks to the Author):

Wu et al. present an updated version of their manuscript presenting Spectroscape as a novel visualization tool for spectral archives.

The authors strongly argue, that their tool must not be viewed as another spectrum clustering algorithm, but a tool to investigate individual spectra.

I believe, that this argument falls short as it is necessary to show that the used method to group spectra compares to existing methods. Nevertheless, I follow the authors argument for this review.

Response:

Thanks for the supportive comments. As we have described in the manuscript, Spectroscape is not a clustering tool. It does not “group spectra” ahead of time, nor does it outputs the list of clusters of the entire spectral archive. So we struggle to see how we can show a comparison of how Spectroscape “group spectra” as compared to existing methods (e.g., MS-Cluster, MaRaCluster, GLEAMS, etc.). Rather, we must emphasize that Spectroscape is a retrieval and visualization tool. It will respond to each individual query spectra (ignoring precursor information), find the nearest neighbors, and compute the connectivity graph among the neighbors on the fly. It is then up to the user to interpret such “local” cluster information. (Of course, this retrieval algorithm can be leveraged to compute the global adjacency matrix efficiently, but that is not the intended application of Spectroscape. Even if we have the adjacency matrix, it would still require a subsequent step to determine the cluster boundaries in an unsupervised manner, before one can compare to the outputs of other clustering tools. This unsupervised clustering function is not part of Spectroscape.) In our view, the performance evaluation of Spectroscape should be in terms of query retrieval, not clustering outcome. We provided evidence to show that the indexing and retrieval method of Spectroscape provides for very high recall, and that the index was designed effectively such that the distances between the 17-byte addresses are well-correlated with the true distances of the corresponding spectra.

I am surprised to see that all benchmarks etc were put in the supplementary methods. Therefore, the manuscript itself holds very limited information that would be relevant to a wider audience.

Response:

Thanks for this constructive comment. Actually, we put them into the Supplementary Information to adhere to the typical length of Nature Communication articles. We felt that the technical details are perhaps only of interest to the real computational experts, which may be a minority of our audience. In any case, we have modified the manuscript to include the comparison to GLEAMS (in terms of approximate nearest neighbor retrieval) in the main text.

1) Novelty

The idea to create a visualization tool on top of a spectrum grouping / ANN network is nice, but not novel. This was done as part of previous PRIDE associated resources (although static and not scalable).

Response:

Thanks for this insightful comment. Yes, we agree that providing a visual representation of clusters is not in itself novel. (We were unable to find the PRIDE cluster visualization alluded to by the reviewer, however.) On the other hand, as the reviewer recognized, Spectroscape is dynamic and scalable, unlike previous tools. Spectroscape did not cluster at all before the neighboring spectra are retrieved. What Spectroscape enabled is the ability to see the neighborhood of a query spectrum, in real time, without any prior offline clustering. Besides, it does not use the precursor m/z to filter for neighbors, meaning that it is effectively performing an “open” search. Since it does not rely on peptide annotations to “train” the index, it is more general

and adaptable to other fields in mass spectrometry. The calculation of the detailed cluster structure (by complete pair-wise similarity calculations among retrieved neighbors) and its display using a force-directed graph, is also unique to Spectroscape.

The methods used by the authors also were introduced by others before. Adapting the training process is innovative, but something I consider incremental improvement. Therefore, overall, the novelty of the proposed work is in my opinion limited.

Response:

We would respectfully disagree with this assessment. Since no example of such similar work was mentioned, we were unable to offer our rebuttal in detail. However, please allow us to state our case in general terms here.

Spectroscape's underlying algorithm, IVF-PQ, was developed by others (Facebook), but the adaptation of it to proteomics in our hands is no doubt novel. (Adapting an algorithm developed by others has also enabled many recent advances in proteomics, e.g., applications of deep learning.) It is important to emphasize that the FAISS library provides many indexing algorithms, of which IVF-PQ is only one of many. It is unfair to suggest that any work that uses the FAISS library is not novel because someone else has used the FAISS library before. On the contrary, precisely because there are so many possible indexing algorithms to choose from, the selection of the specific indexing algorithm and its adaptation and optimization for the task at hand is not trivial.

Perhaps the reviewer was taking a broader view that any method that relies on some dimensionality reduction to speed up the search of neighbors in high-dimensional space should be considered "similar" and therefore not novel or merely "incremental." But if this is the case, the novelty bar is set unreasonably high. For example, we would argue that GLEAMS, which was published recently in Nature Methods, also adopts a similar conceptual approach. Its novelty, as in our case, lies in the choice of the exact dimensionality reduction algorithm (i.e., in how the "embedding" was trained) to suit its particular application. Moreover, we have shown that the approximate nearest neighbor retrieval of Spectroscape is more effective and general than GLEAMS's, not to mention that Spectroscape enables real-time and dynamic visualization of the complete cluster structure among neighbors, a feature that no other tool can boast.

2) Software

The state of the software was improved but is still below expectations. The installation works. Nevertheless, building an archive with only one 750 MB mzXML file failed with a segmentation fault on a 16 core machine with 32 GB RAM (again using a docker container and ubuntu:22.04). The mzXML file was taken from PXD016673.

Response:

Thanks for pointing out this issue. We apologize for forgetting to mention that a minimum of 100,000 spectra is required to initialize the Spectroscopie index. One mzXML file is not sufficient for the k-means algorithms to initialize the “buckets”. In our demo, we use 24 mzXML files, and we do not observe the error. We have updated the user manual to explain this, and provide a clear error message if the user fails to provide enough training data. We also uploaded a test dataset so that users can try out the tool more easily.

The website to test the live version at <http://omics.ust.hk:8709/index.html> was not accessible during the time of the review (tested from multiple networks).

Response:

Thanks for pointing out this issue. Our server was down for several days during the time of review due to electricity suspension and hardware upgrade in our institution, which was out of our control. (We did make an announcement on GitHub around that time to warn potential users; nonetheless, we fully accept that it is our responsibility to keep it running.) To make the web demo works more persistently, we recently purchased an domain name spectroscopie.cc and a server instance in AWS, which connects to multiple mirror servers to provide a bit more redundancy. We are confident that our new web interface (spectroscopie.cc) will always be online during the period of the review and beyond.

Overall, I therefore have severe reservations whether this tool will provide the simple and high usage of spectrum clustering or similar methods. It also raises grave concerns whether the authors hold the required expertise to develop an end-user ready piece of software and maintain it for a relevant amount of time. The original version contained severe and obvious errors and highlighted that the software was not tested properly. Apparently, the software still has not reached a stable version.

Response:

We again apologize for the web interface issues, which are now fixed. We will continue to maintain the software and ensure its usability.

3) Theoretical usability

The tool (if it works) will only serve one question in proteomics research, to validate specific PSMs. The authors are correct, that some findings center on specific proteoforms / PTMs. Nevertheless, a vast number of literature centers on large-scale characterisations of proteomics samples. These will not profit from this tool.

Response:

Yes, we agree that the large-scale characterization of proteomics samples will always be important. However, we do not lack tools (e.g. database search engines) for that purpose. Spectroscopie serves a different purpose: for validating biological findings that critically depend on the identification/quantification of specific genes/proteins (which could be discovered by database search engines initially). Clearly, the credibility of a single PSM or a few selected PSMs cannot be established by current approaches of global FDR control approach at the dataset level. Spectroscopie helps in the validation of such findings with information that is not readily accessible to the researcher now. Spectroscopie is therefore complementary to, and not competing with, existing tools for high-throughput characterization. We have added a paragraph in the revision to clarify this point.

As a mere proof of concept, this work is interesting, but not to the wide community addressed by this Journal. I therefore suggest this work to be submitted to a field-specific one.

Response:

We do not dispute that this is a proof of concept at this stage, but Spectroscopie is a one-of-a-kind addition to our toolkit that may have long-lasting impact to our field. In the manuscript we have articulated a vision of how proteomics data analysis should be more collaborative and better integrated with data sharing, with Spectroscopie being the interface between the user and the repository. Although this will require the active participation of many stakeholders and cannot be realized now, we believe that getting the word out is the critical first step. Of course, whether this step is “big” enough for this journal is a question for the editorial leadership of Nature Communications. On the other hand, we have also seen many publications in journals with broad readership that are proof of concept in nature, or target audience in a specific field exclusively. In our field, for example, we can think of many recent examples, such as MSFragger (2017 Nat Methods), DeepNovo (2017 PNAS), Prosit (2019 Nat Methods), AlphaPeptDeep (2022 Nat Comm), GLEAMS (2022 Nat Methods), etc.

Reviewer #3 (Remarks to the Author):

The authors have clarified the scope of the current work and have offered a compelling defense of the current capabilities of Spectroscopie. There will be researchers looking at novel proteoforms or other non-standard identifications, and Spectroscopie provides a unique tool for quickly exploring possible hypotheses regarding these discoveries. Additionally, the new analyses provided by the authors go a long way towards suggesting future developments for the technology.

Response:

Thank you very much for the supportive comments. We believe the Spectroscopie tool has the potential to shed light on the unidentified spectra and verify new identifications against the massive amount of existing data, which we cannot easily do right now.

In their rebuttal, the authors offered the following argument in defense of Spectroscape:

We believe that as a field we have gone too far down the road of controlling errors at a global level (e.g., by estimating FDR by target-decoy searching), while side-stepping the more fundamental question of what makes an individual PSM believable in the first place. Automatic “black box” approach to validation appeals to researchers who would rather focus on the biology, but we pay for this convenience with our credibility in the eyes of other life scientists. In a perfect world, the global error control we do today should only be viewed as the first step to filter away unpromising leads, but we should not be content to accept everything that passes the filter as truth. In just about every other discipline, scientists would not embark on expensive independent validation experiments until technical validation is done first and the data is deemed trustworthy. In proteomics, many researchers do not bother to even look at a spectrum that underpins our hypothesis, but are all too willing to embark on expensive and tedious biological validation experiments. With the right tool, checking even hundreds of PSMs perhaps take a matter of hours; in contrast, biological validation experiments often take months.

I would suggest adding something along these lines to the main manuscript. The argument against global FDR control is not new. Many papers have been written about the differences between global FDR and local FDR or the False Positive Rate of a spectrum. For example, <https://www.ncbi.nlm.nih.gov/pmc/articles/PMC2689316/>. These issues take on added importance when discovering novel proteoforms or searching non-standard databases. Even simple phospho enrichment experiments typically have different empirical FDR rates when subsetting the data to modifications on tyrosine. Referencing the known failures of global FDR control would bolster the argument in favor of using Spectroscape as a validation step.

Response:

Thanks for the suggestions. We have modified the main text to include discussion of how global FDR control is not sufficient to establish the credibility of a small subset of the data, and why the ability to validate individual PSMs will always have an important place in proteomics.

Regarding the main manuscript, my only new suggestion is that it would be nice for references to the Supplement to include a specific section identifier. In the current form it is somewhat inconvenient to match the main text with the relevant section of the supplement.

Response:

Thanks for the suggestion. We have assigned specific identifiers to different sections of the Supplementary Information.